# Undergraduate Students’ Online Health Information-Seeking Behavior during the COVID-19 Pandemic

**DOI:** 10.3390/ijerph182413250

**Published:** 2021-12-16

**Authors:** Wan-Chen Hsu

**Affiliations:** Center for Teaching & Learning Development, National Kaohsiung University of Sciences and Technology, No. 1, University Rd., Yanchao Dist., Kaohsiung City 824, Taiwan; wanchen@nkust.edu.tw; Tel.: +886-7-3814526 (ext. 31171)

**Keywords:** health information, online health information, information-seeking behavior, undergraduate students, COVID-19

## Abstract

As the COVID-19 pandemic has swept across the world, the amount of health-related information available has skyrocketed. Individuals can easily access health information through the internet, which may influence their thoughts or behavior, causing potential technological risks that may affect their lives. This study examined the online health information-seeking behavior of undergraduate students. Taking health issues as a guiding framework, content analysis was adopted to assess participants’ online health information-seeking behavior using a computer screen recording software, and coding analysis was conducted. The study was conducted during the COVID-19 pandemic with a formal sample of 101 participants. In terms of online health information-seeking behavior, 59% of the study participants used nouns as keywords, only 27% used Boolean logic retrieval techniques, 81% paid attention to the date of the data, and 85% did not consider the author’s professionalism. The results indicate that health information-seeking behavior and outcome judgments may be a missing piece of the puzzle in higher education. Consequently, the development of online health information-seeking skills through programs for undergraduate students is suggested to ensure that online readers have access to appropriate health information.

## 1. Introduction

The COVID-19 pandemic, caused by the novel Sars-CoV-2 coronavirus, has caused the death of millions of people and disrupted daily life worldwide. During this pandemic, individuals were restricted from going outside, and physical activities were reduced as a result of its impact. Consequently, people gathered, exchanged information, and entertained themselves via the internet, with online health information becoming an alternative to personal visits to physical hospitals and medical centers. On 19 May 2021, Taiwan’s Ministry of Education announced that students at all levels would stop attending schools, fully initiating online instruction. The impact of campus closures and significant social changes brought many challenges to higher education, affecting personal internet use. The COVID-19 pandemic brought into renewed focus the health of students in higher education, which already necessitated concern [1]. In Taiwan, the internet is the source of health-related information for 100% of undergraduate students who avail it on a frequent basis [2]. When these students encounter health-related problems, they often resort to the internet to obtain information as a temporary solution [3].

Online platforms have the potential to provide individuals with useful information, increase their engagement, and potentially revolutionize the patient–physician relationship [4]. Information seeking has become a focus of health communication scholarship, since individuals can now use a variety of platforms, such as the television, newspapers, the internet, and other interpersonal communication channels, to gain knowledge [5]. Chen and Lee [6] noted that people often have limited skills related to retrieving and evaluating the vast amount of information available from a variety of online sources with varying quality. This overwhelming availability of online health information highlights the importance of understanding the status and key influencing factors of its use among individuals.

Health information is defined as information that can assist individuals in promoting their health, making health-related decisions, and participating in the healthcare system [7]. Information seeking can be unintentional, passive, or active [8] and is often purposeful, with individuals seeking information to meet a personal need or goal [9]. Information-seeking behavior is the action of searching for and using information in any way, following an individual’s need. In particular, it relates to the behavior arising from an interaction with the information source when one needs information; it can range from passive attention to passive searching, active searching, and ongoing searches, all of which fall within the scope of information-seeking behavior [9]. Online health information-seeking behavior is dominated by active information seeking and passive information acquisition [7]. Health information-seeking behavior is a type of personal health promotion in which individuals obtain expertise from various sources, such as doctors, to inform their decisions, improve their food and nutrition intake, relieve stress, and reduce drug abuse [5]. In sum, online health information-seeking behavior involves individuals’ retrieval of health information from the internet, which can be actively or passively motivated, for the purpose of obtaining knowledge for personal health promotion and facilitating decision making.

Regarding health information retrieval and health promotion theories, the social cognitive theory is one of the most widely used theoretical frameworks [10]. Bandura’s social cognitive theory provides a structure for interpreting the relevant results of individuals after retrieving information [11,12]. For example, how much confidence an individual has in finding quality health information, i.e., their self-efficacy in searching, is also related to the expected results after retrieval. Self-efficacy can be a powerful predictor of expected results regarding an individual’s online health information-seeking behavior [13]. The risk information seeking and processing model (RISP) is one of the representative theoretical models explaining online information seeking. It emphasizes that the behavior of individuals to retrieve online information is triggered by insufficient cognitive data (termed as information insufficiency hereafter); according to the model, a lack of information is the main factor directly driving information seeking, alongside other incidental social and psychological factors, such as emotional response (worry, anxiety) and subjective criticism of information. The RISP model thus provides a framework to explain the key influencing factors that individuals use to seek and process relevant risk information in a more systematic or deliberate manner. Brown, Skelly, and Chew-Graham [14] proposed a model, pointing out that individuals’ online health information retrieval is affected by their previous experience, health beliefs, and other personal background factors.

Research on health information-seeking behavior in Taiwan remains in its infancy. Previous studies have focused on the content of health information texts [15,16], the effect of health information on readers’ intention to use it [17], health information-seeking experiences [17,18], the relationship between online information seeking and cognitive factors [19], and how post-search emotions affect social cognitive factors and perceptions, indirectly shaping attitudes and behavior [13]. Information literacy, one of the core competencies of eHealth literacy, is an individual’s ability to understand how to effectively search for, organize, and use information, for example, by retrieving relevant information using a keyword [20].

Health-related issues, such as health literacy, are more frequently discussed in the context of adult health decision making and health behavior. Although adolescents need to increase their sense of responsibility for maintaining their own health, less research has been conducted among this age group [21]. The present study, therefore, investigated the online health information-seeking behaviors of undergraduate students, using common health problems as a guide. Here, online health information-seeking behavior was defined as individuals retrieving health information through the internet. The specific behavioral items observed were “keyword selection”, “information browsing”, and “information sources”; suggestions were then devised for a skills development program to shape undergraduate students’ online health information-seeking behaviors based on the findings. The academic contributions of this study could enrich our knowledge and theoretical scope of online health information-seeking behavior issues, highlighting their value for students in the COVID-19 era.

## 2. Materials and Methods

### 2.1. Study Methods

This study examined undergraduate students’ online health information-seeking behavior during the COVID-19 pandemic. In this study, the Delphi method was used for gaining consensus through controlled feedback from a panel—a group made up of experts in the subject. The method is often used when there is limited or conflicting evidence, the participants may be geographically dispersed, and anonymity is desired to control for dominant individuals. The Delphi method consists of panel selection, the development of content surveys, and iterative stages of anonymous responses to gain consensus [22]. The relevance and objectives of Delphi techniques differ among various disciplines. While they are primarily used in the context of technical and natural sciences to analyze future developments, they are also used in health sciences to reach consensus [23].

In the initial stage of this study, test questions on the health issues sought by college students online were developed. The team members tasked with the development of these questions included scholars and experts in the fields of health promotion and hygiene education, education testing, and physicians and nurses with rich experience in medical services. The investigation was continued with the research questions on health problems sought online by college students.

We conducted interview surveys of 101 students from four universities to understand their online health information-seeking behavior. The data were analyzed using both quantitative and qualitative methods. Content analysis is a research tool that is used to determine the presence of certain words, themes, or concepts within qualitative data [24]. In this study, content analysis was used to pre-program health questions and solicit undergraduate students from four universities in south and central Taiwan to participate. Prior to data collection, student participants gave their consent to be profiled in an online retrieval behavior video and were asked to find appropriate answers to health-related questions on the internet. The video data were then coded and analyzed to understand the status of online health information-seeking behaviors demonstrated by undergraduate students.

### 2.2. Study Participants

The sample consisted of students on campus who voluntarily wished to participate. Data were collected during the COVID-19 pandemic between March and May 2020 from public libraries on the campuses of the four universities. It was ensured that the participants’ privacy was protected and that they would not be disturbed during participation. The students agreed to use a browser to search for information pertaining to the preset health questions and have their screens recorded during the process. The final sample comprised 31 students from one university in central Taiwan and 70 students from three universities in southern Taiwan—resulting in a total sample size of 101. A total of 101 valid responses were thus obtained for image content analysis.

### 2.3. Study Tools

To investigate the online health information-seeking behavior of the undergraduate students, this study referred to the “14 Health Topics of the Health Promotion Administration, Ministry of Health and Welfare (Taiwan)”, the “Top 10 Health Education and Teaching Issues in the United States”, and the six categories of health information in Liao et al.’s [7] study, as the basis for formulating the example health issues. Liao et al.’s categories included disease treatment, diet and nutrition, exercise and fitness, health and aging prevention, medical consultation and treatment, and preventive health care, being supplemented with health issues of public concern. The health issues of concern to undergraduate students in this study comprised four topics: “balanced diets”, “obesity prevention”, “health and fitness promotion”, and “sleep management”, each of which was extended to two questions, for a total of eight questions. The students were asked to select one of the eight questions and provide written answers to it, in order to reveal the status of their online health information-seeking behaviors and actual behaviors. Taking “balanced diet” as an example, the design concept of the health-related questions used in the research is shown in Table 1.

### 2.4. Data Collection and Analysis

The study was ethically reviewed according to the human research ethics governance framework, and participants were asked to complete an informed consent form prior to data collection. An oCam screen recording program was used to record the online health information retrieval behavior of each participant on the computer screen, which was transcribed for coding and analysis following the study’s completion. The content analysis framework of the participants’ online health information retrieval behavior included “keyword selection”, “Boolean logic query”, “limited scope for query”, “information browsing”, and “information source”.

The reliability of the content analysis was measured using inter-rater reliability [26], in which higher consistency results indicate higher reliability of the analysis. The reliability coefficients for the coding results were calculated according to the formulae of mutual agreement, mean agreement, and reliability coefficients, as follows
$$\mathrm{Mean}\;\mathrm{agreement}=\frac{2M}{N1+N2}$$
$$\mathrm{Reliability}\;\mathrm{coefficient}=\frac{n\times Mean\;mutual\;agreement\;}{1+\left(n-1\right)\times Mean\;agreement}$$

M: The number of variables for which the coding result was fully agreed between two persons.

N1: The total number of variables coded by the first coder.

N2: The total number of variables coded by the second coder.

n: The number of coders.

Content analysis of the data could only be performed following the determination of the reliability coefficient. Two coders coded 30 samples and calculated a mean agreement of 0.67 and a reliability coefficient of 0.80.

## 3. Results

In this study, the Delphi method was used to encode the data content of the responses to the survey results, using descriptive statistics to restore the current status of college students’ online information retrieval behavior.

### 3.1. Undergraduate Students’ Online Health Information-Seeking Behavior Is Mostly Based on Using Nouns as Keywords, with Few Using Boolean Logic Techniques, and Unlimited Scope for Queries

A skillful use of internet search functions, such as the selection of keywords, application of Boolean logic, and limitation of the query scope, allows users to focus more specifically on the relevant online information during the search process, filtering out unnecessary information. Table 2 shows the status of online health information-seeking behaviors among the undergraduate students surveyed, as well as their actual behaviors. When choosing keywords for their searches, 59% of the participants used nouns as keywords; 43% used nouns, adjectives, and adverbs as common keywords; and 28% used sentences. Regarding the search technique of Boolean logic, only 27% of the participants used the operators “AND”, “OR”, and “NOT”. In terms of limiting the scope of their queries, 12% of the study participants limited the type of data searched, while only 2% limited the date and language of the data retrieved; this indicated a low percentage of users who limit the date, form, and language of information for narrowing down the scope of their searches.

### 3.2. Status of Online Health Information-Seeking Behavior among Undergraduate Students: Information Browsing and Information Sources

The results showed that the average number of web pages visited by the study participants to determine the adequacy of the information available on a given health topic was 2.99; their overall browsing time was 5.54 min; and the average time they spent on each web page was 2.39 min. Regarding the information source, 81% of the respondents were concerned about the newness of the information and the year of publication. The information sources consulted were mostly “organization websites” (45%) and magazines or periodicals (40%), while news reports (8%), forums and chat rooms (13%), and personal websites (22%) accounted for a minority of the information sources. However, in terms of the professionalism of the data sources, 22% of the users believed that the authors of the data they retrieved were experts in the related fields, and 42% of the data mentioned the author’s affiliation; however, 85% of the users found that the authors of the data were anonymous, or believed that they were unprofessional, as shown in Table 3.

## 4. Discussion

### 4.1. Is Information Literacy the Missing Part of Health Promotion among Undergraduate Students?

It was found that most of the keywords used by the participants in the search for health information were nouns, although some did use a mixture of nouns, adjectives, and adverbs. Few searched using Boolean logic, and they seldom limited the scope of their queries to narrow down the results, indicating that the undergraduate students had few relevant skills in searching for information.

Information literacy is one of the multiple components of health literacy that adolescents are aware of, encompassing a range of skills and knowledge that are relevant to health behaviors and can reduce health risks [21]. When individuals are familiar with internet search methods, they can easily filter out useful information based on the purpose of the search and the source of the data. Conversely, users who are unfamiliar with these operations are easily distracted by irrelevant information, which affects the accuracy and usefulness of their information judgments. Furthermore, individuals who are exposed to a large amount of online health information and are unable to critique and make good use of this information may suffer negative effects, leading to feelings of anxiety that can cause emotional distress and even severe cyberchondria [27]. Joseph and Fleary [21] explored adolescents’ perceptions of health literacy and revealed that they involved more functional than critical literacy. Criticality involves reading, understanding, and acting upon health information, having potential effects and benefits for individuals and society. This highlights the importance of critical skill development and education for the youth in particular.

### 4.2. What Is the Potential Risk of Self-Diagnosis Due to the Explosion of Health Information during the COVID-19 Pandemic?

This study found that 81% of the participants were concerned about the newness of the information they found and the year of the source. In terms of the professionalism of the source, 22% of the users believed that the author of the information they retrieved was an expert in the field. Meanwhile, 42% of the information retrieved mentioned the author’s affiliation. However, 85% of the participants were dubious about the information they found on the internet, as its author was either anonymous, or they believed the author was not a professional. In fact, obtaining health-related information on the internet and diagnosing oneself based on it affects one’s health-related behaviors, decisions, and actions. Sturiale et al. [4] found that there was a correlation between those who used the internet for work and those who had knowledge of both symptoms and their likely diagnosis before consultation, among patients. Patients who used the internet daily were more likely to request a consultation within six months of symptom onset. Additionally, those with anorectal diseases were more likely to have knowledge of their disease and symptoms before the visit. Hsu et al. [3] surveyed a sample of undergraduate students to explore their experiences with online health information and found that they retrieved health information related to their needs from the internet in order to prevent or maintain their health conditions. However, the prescriptions they retrieved online only offered reference answers, and sometimes inner doubts still lingered in their minds. Using the flu as an example, Myrick employed a naturalistic experiment to test the emotions of 380 Americans after retrieving information online, exploring the theoretical models that shaped cognition and behavior [13]. It was found that the study participants had difficulty retrieving information when they had a dubious attitude. Myrick further tested how to improve the skills required for the online health information retrieval process, observing that individuals had multiple emotions (fear, hope, satisfaction, interest, and motivation) after retrieving information, and the mediating effect of “social cognitive factors” affected their subsequent attitudes and behaviors. The positive emotions of interest and hope experienced during the online health information-seeking process positively influenced individuals’ confidence and behavioral intentions.

The number of medical articles published on the internet increased significantly during the COVID-19 pandemic [28]; however, at the same time, the amount of fake news and disinformation skyrocketed to several dozen times the previous level [29]. As the internet booms and health information spreads, the World Wide Web has become a major source for the public to search for information about medical and health risks. In tandem with this boom, many health and disease-focused websites have emerged to provide the public with more immediate access to health information. Such sites provide information and resources for readers with medical conditions, assisting them with possible self-diagnostic references for certain symptoms and helping them decide whether to self-treat or consult a physician [30]. The use of the internet to retrieve health-related information is a behavioral manifestation of the individuals’ search for peace of mind. However, the information available on the internet is not always accurate and reliable; therefore, it is important to promote individuals’ online search skills to reduce uncertainty, worries, and anxiety, avoiding incorrect self-diagnosis. As individuals are exposed to the risks of online information technology, it is critical to understand how they use health information when they are inundated with it online [31]. A key strategy for managing health care surge is “forward triage”—the sorting of patients before they arrive at the emergency department (ED). Direct-to-consumer (or on-demand) telemedicine, a 21st-century approach to forward triage that allows individuals to be efficiently screened, is both patient-centered and conducive to self-quarantine, protecting patients, clinicians, and the community from exposure to any infectious disease, such as COVID-19. Furthermore, it allows physicians and patients to communicate using smartphones or webcam-enabled computers, which may be beneficial during situations such as the COVID-19 pandemic [32]. Telemedicine, however, may not always be the go-to approach for physicians in Italy. For example, the utilization of telemedicine for the diagnosis of common proctologic conditions (e.g., hemorrhoidal disease, anal abscess and fistula, anal condylomas, and anal fissure) and functional pelvic floor disorders was generally considered inappropriate. Teleconsultation was instead deemed appropriate only for the diagnosis and management of pilonidal disease, revealing the boundaries of telemedicine in Italy. Therefore, infrastructures, logistics, and legality related to telemedicine need to be standardized [33].

## 5. Conclusions

This study investigated the online health information-seeking behavior of undergraduate students. The results revealed that most of the keywords used by the study participants when searching for health information were nouns, although some used a mixture of nouns, adjectives, and adverbs. Few participants searched using Boolean logic, and few limited the scope of their queries to narrow down the retrieved data. Almost all the study participants questioned the validity of the information they found, considered the authors of the data to be anonymous or non-professionals, and were dubious about the information available on the internet.

The widespread availability of e-health information has become an important issue for public health gains. From the viewpoint of the reader, individuals are exposed to a large amount of information that is easily accessible for everyone on the internet, suggesting that technological risks are relevant to individuals’ lives but are often widely ignored or overlooked. It is suggested that in the future, the online health information retrieval skills needed by adolescents can be appropriately integrated into university curricula in the form of training through relevant information collection skills and expertise, such as clinical understanding, prevention strategies, and navigation of the healthcare system [27]. Students’ skills in searching for information and their ability to distinguish between true and false information should also be fostered.

This study had a few limitations. The Delphi method used in the research has its own restriction, such as the identification of “consensus” amongst experts, which appears to be the central motivation for the application of Delphi techniques in health sciences. Nevertheless, there is no general definition for what consensus actually is. As far as the research replicability is concerned, this study was aimed at college students, and there were limitations related to the ecological validity of our research results due to the small sample size. Future studies should, therefore, employ larger research samples, using this article as an introduction for further analysis regarding the process of seeking health information in relation to the COVID-19 pandemic. In terms of the study design, this research asked respondents to answer pre-designed health questions, which may have limited its intrinsic validity, failing to assess the online health information retrieval behavior of individuals when they face personal health problems. In addition to designing a series of health questions to explore the participants’ online health information retrieval practices through observational methods, future research could ask participants to describe their online health information retrieval process in a “think aloud” manner to better understand their subjective use.

## Figures and Tables

**Table 1 ijerph-18-13250-t001:** Example of the design concept of health-related questions of concern to undergraduate students.

Health Question Design	Reason and Reference for Health Question Design ^1^	Correct Answer Reference ^1^
The slogan “Five Servings of Fruit and Vegetables a Day” encourages people to eat five servings of fruit and vegetables every day. If you eat five servings of the recommended weight of vegetables in a day, how many grams of vegetables do you think you should eat?	According to the Health Promotion Administration’s Health Behavioral Risk Factor Surveillance System (BRFSS) 2016, only 12.9% of adults aged 18 or above (9.4% of men and 16.3% of women) met the recommended daily intake of three servings of vegetables and two servings of fruit, which was less than the recommended number of servings in the Dietary Guidelines. Only 20.7% of the surveyed citizens consumed five servings of fruit and vegetables.	The Health Promotion Administration reminds the public to develop a healthy diet that includes “three servings of vegetables and two servings of fruit”, by consuming three servings of vegetables (one serving of cooked vegetables is about half a bowl) and two servings of fruit (one serving of fruit is about the size of a fist) every day, and to select local, seasonal, colorful vegetables and fruits in their original state.

^1^ Data source: [25].

**Table 2 ijerph-18-13250-t002:** Current status of health information retrieval behavior among undergraduate students: Information seeking.

Check Questions about Online Health Information Retrieval Behavior		Code Type	
Keyword selection	1. How to use keywords: Nouns as keywords	Used 60 (59%)	Not used 41 (41%)
	2. How to use keywords: Nouns, adjectives, and adverbs as common keywords	Used 43 (43%)	Not used 58 (57%)
	3. How to use keywords: Sentences as keywords	Used 28 (28%)	Not used 73 (72%)
Boolean logic query	How to reduce the scope of data and whether to use the operators “AND”, “OR”, and “NOT”	Used 27 (27%)	Not used 74(73%)
Unlimited scope for query	1. Whether to limit the scope of the query: Limit the date of the unnamed title	Yes 2 (2%)	No 99 (98%)
	2. Whether to restrict the scope of the query: Limit the data type	Yes 12 (12%)	No 89 (88%)
	3. Whether to limit the scope of the query: Limit the language	Yes 2 (2%)	No 99 (98%)

**Table 3 ijerph-18-13250-t003:** Status of health information retrieval behavior among undergraduate students: Information browsing and information sources.

Check Questions about Online Health Information Retrieval Behavior		Code Type	
Information browsing	1. Number of pages viewed	Min. 1, Max. 14, Mean 2.99 (SD = 2.33)	
	2. Overall query time	Min. 1.42 s, Max. 17.43 s, Mean 5.54 s (SD = 3.04)	
	3. Average time on page	Min. 0.65 s, Max. 6.23 min, Mean 2.39 min (SD = 1.31)	
Information source	1. Whether the data source is new or old (year) can be marked in the Word file	Directly expressed in AD year 82 (81%)	No time for data 19 (19%)
	2. Whether there is a source of information: Website of the organization	Yes 45 (45%)	No 56 (55%)
	3. Whether there is a source of information: Magazines or periodicals	Yes 40 (40%)	No 61 (60%)
	4. Whether there is a source of information: News reports	Yes 8 (8%)	No 93 (92%)
	5. Whether there is a source of information: Forums or chat rooms	Yes 13 (13%)	No 88 (87%)
	6. Whether there is a source of information: Related research papers	Yes 0 (0%)	No 101 (100%)
	7. Whether there is a source of information: Personal web pages	Yes 22 (22%)	No 79 (78%)
	8. Professionalism of the information source: The author is a professional, for example, an expert in a related field or a physician	Yes 22 (22%)	No 79 (79%)
	9. Professionalism of the information source: The author is a professional, and their affiliation is mentioned	Yes 42 (42%)	No 59 (58%)
	10. Professionalism of the information source: The author of the data is anonymous or a non-professional	Yes 85 (84%)	No 16 (16%)

## Data Availability

Not available.

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
