# Peer review of "Undergraduate Students’ Online Health Information-Seeking Behavior during the COVID-19 Pandemic"

_ijerph, 2021, doi:10.3390/ijerph182413250_

Round 1
Reviewer 1 Report
The proposal is attractive. However, to improve, you need the next questions:
-Theoretical framework: to update some references. For example, there are references of 2001, 2006…
-Methods. You can complete Methodology with a qualitative tool (Delphi, in this case), because, if not, it is only quantitative.
-Results: it will be improved with the methodological new tools.
-Discussion: improvable with the qualitative new data.
Author Response
Dear Editor:
Thank you for recommending my manuscript for publication, subject to revision. The manuscript has been revised in accordance with the reviewers’ constructive comments (see the annotated table below). A number of specific comments and editorial suggestions that were offered by the reviewers and the editor have been incorporated and addressed in the revised version of the manuscript. In addition, professional editors have edited the paper.
Again, I thank you for kindly offering me this opportunity. I sincerely hope that the revised manuscript is suitable for publication in the International Journal of Environmental Research and Public Health. Your comments are greatly appreciated, and I look forward to your reply.
Regards,
…..
Wan-Chen Hsu
|
Reviewer’s comments |
Reply |
|
Reviewer 1 |
|
|
1. The proposal is attractive. However, to improve, you need the next questions: -Theoretical framework: to update some references. For example, there are references of 2001, 2006… |
Thank you for the comments. The author has updated the references and added the following references.
|
|
2. -Methods. You can complete Methodology with a qualitative tool (Delphi, in this case), because, if not, it is only quantitative. |
Thank you for the comment. The author has revised and clarified the statement.
We conducted interview surveys of 101 students from four universities to understand the online health information-seeking behavior. The data were analyzed through quantitative and qualitative methods. In this study, content analysis was used to pre-program health questions and to solicit undergraduate students from three universities in south and central Taiwan to participate in the study.
|
|
3. -Results: it will be improved with the methodological new tools. |
Thank you for the comment. The author has revised and clarified the statement.
In this study, a content analysis method was used. To begin conceptual content analysis, first, the research question must be identified and the sample for analysis must be selected. Next, the text must be coded into manageable content categories—which is basically the process of selective reduction. By reducing the text to categories, the researcher can focus on and code for specific words or patterns that inform the research question. |
|
4. -Discussion: improvable with the qualitative new data. |
Thank you for the comment. The author has revised and clarified the statement.
In this study, a content analysis method was used. To begin conceptual content analysis, first, the research question must be identified and the sample for analysis must be selected. Next, the text must be coded into manageable content categories—which is basically the process of selective reduction. By reducing the text to categories, the researcher can focus on and code for specific words or patterns that inform the research question. |
Reviewer 2 Report
This is an original article regarding Undergraduate Students' Online Health Information-Seeking behavior during COVID-19 Pandemic
The topic is promising.
The methodology needs to be improved. Some references are missing in the text ([])
How were participants involved? On the basis of what were they chosen? When was the study conducted? Did everyone who participated answered?
Why was an online delivering system such as survey monkey or other not used?
Why hasn't a particular type of information been indicated?
I strongly suggest to include and discuss the following articles:
- Internet and social media use among patients with colorectal diseases (ISMAEL): a nationwide survey. Colorectal Dis. 2020 Nov;22(11):1724-1733. doi: 10.1111/codi.15245
- Effects of COVID-19 on College Students' Mental Health in the United States: Interview Survey Study. J Med Internet Res. 2020 Sep 3;22(9):e21279. doi: 10.2196/21279
Maybe a chapter regarding the future perspective can be useful
Telemedicine?
Virtually Perfect? Telemedicine for Covid-19. N Engl J Med. 2020 Apr 30;382(18):1679-1681. doi: 10.1056/NEJMp2003539
E-consensus on telemedicine in proctology: A RAND/UCLA-modified study. Surgery. 2021 Aug;170(2):405-411. doi: 10.1016/j.surg.2021.01.049
English-language editing (mild)
Author Response
Dear Editor:
Thank you for recommending my manuscript for publication, subject to revision. The manuscript has been revised in accordance with the reviewers’ constructive comments (see the annotated table below). A number of specific comments and editorial suggestions that were offered by the reviewers and the editor have been incorporated and addressed in the revised version of the manuscript. In addition, professional editors have edited the paper.
Again, I thank you for kindly offering me this opportunity. I sincerely hope that the revised manuscript is suitable for publication in the International Journal of Environmental Research and Public Health. Your comments are greatly appreciated, and I look forward to your reply.
Regards,
…..
Wan-Chen Hsu
|
Reviewer’s comments |
Reply |
|
Reviewer 2 |
|
|
1. This is an original article regarding Undergraduate Students' Online Health Information-Seeking behavior during COVID-19 Pandemic The topic is promising. The methodology needs to be improved. Some references are missing in the text ([]) |
Thank you for the comments. Accordingly, the author has revised the methodology and has checked the references and added the following new references.
|
|
2. How were participants involved? On the basis of what were they chosen? When was the study conducted? Did everyone who participated answered? Why hasn't a particular type of information been indicated? |
Thank you for the comments. Accordingly, the author has revised and clarified the statement in methodology.
This study aimed to understand the online health information-seeking behavior of undergraduate students. The participants in the study came from a voluntary sample solicited on-site on campus. Data were collected from public libraries on the campuses of the four universities. It was ensured that the participants’ privacy was protected and that they would not be disturbed during participation. The participants agreed to use a browser to search for information pertaining to the preset health questions and have their screens recorded during the process. Data were collected between March and May 2020. A total of 101 valid responses were obtained for image content analysis.
|
|
3. I strongly suggest to include and discuss the following articles: · Internet and social media use among patients with colorectal diseases (ISMAEL): a nationwide survey. Colorectal Dis. 2020 Nov;22(11):1724-1733. doi: 10.1111/codi.15245 · Effects of COVID-19 on College Students' Mental Health in the United States: Interview Survey Study. J Med Internet Res. 2020 Sep 3;22(9):e21279. doi: 10.2196/21279 Maybe a chapter regarding the future perspective can be useful Telemedicine? Virtually Perfect? Telemedicine for Covid-19. N Engl J Med. 2020 Apr 30;382(18):1679-1681. doi: 10.1056/NEJMp2003539 E-consensus on telemedicine in proctology: A RAND/UCLA-modified study. Surgery. 2021 Aug;170(2):405-411. doi: 10.1016/j.surg.2021.01.049 |
Thank you for the comments. Per your comment, the author has updated the references and added the following seven new references.
|
|
4. English-language editing (mild) |
Thank you for the comment. The author has conducted English-language editing before resubmitted. |
Reviewer 3 Report
The article entitled "Undergraduate Students' Online Health Information-Seeking. Behavior During the COVID-19 Pandemic" deals with a very interesting topic of analyzing the process of seeking health information in relation to the COVID-19 pandemic. The pandemic is also a current and very important social subject, for example, due to the risks that COVID brings. For this reason, it is extremely important to properly inform about the situation and assess it based on acquired data.
The analyzed article has a theoretical and empirical character. It contains several important conclusions and judgments.
Surveys were conducted on a sample of 101 students from 4 universities. The research that the authors used covered students, yet it can be considered that despite such a limited scope and small research sample, this article can serve as an introduction to further analysis and the search for insights in this area. Therefore, the sample can be considered sufficient, but absolutely minimal for similar analyses. At this level, if I were to find errors, I would say that the sample that the authors used could be expanded. Therefore, in my opinion, the results of the obtained research cannot be generalized. In my opinion, such information should also be included in the methodology and the summary.
After supplementation, I believe that the article can be published. However, it is crucial to point out that the sample on which it was based can be treated more as a poll.
Author Response
Dear Editor:
Thank you for recommending my paper for publication as subject to revision. The paper has been revised according to the reviewers’ constructive comments (see the annotated table below). A number of specific comments and editorial suggestions that were offered by the reviewers and the editor have been incorporated and addressed in the revised paper. In addition, professional editors have edited the paper.
Again, I thank you for kindly offering me this opportunity. I sincerely hope the revised paper is suitable for publication in International Journal of Environmental Research and Public Health. Your comments are greatly appreciated, and I look forward to your reply.
Regards
…..
Wan-Chen Hsu
|
Reviewer’s comments |
Reply |
|
Reviewer 3 |
|
|
The article entitled "Undergraduate Students' Online Health Information-Seeking. Behavior During the COVID-19 Pandemic" deals with a very interesting topic of analyzing the process of seeking health information in relation to the COVID-19 pandemic. The pandemic is also a current and very important social subject, for example, due to the risks that COVID brings. For this reason, it is extremely important to properly inform about the situation and assess it based on acquired data.
The analyzed article has a theoretical and empirical character. It contains several important conclusions and judgments.
Surveys were conducted on a sample of 101 students from 4 universities. The research that the authors used covered students, yet it can be considered that despite such a limited scope and small research sample, this article can serve as an introduction to further analysis and the search for insights in this area. Therefore, the sample can be considered sufficient, but absolutely minimal for similar analyses. At this level, if I were to find errors, I would say that the sample that the authors used could be expanded. Therefore, in my opinion, the results of the obtained research cannot be generalized. In my opinion, such information should also be included in the methodology and the summary.
After supplementation, I believe that the article can be published. However, it is crucial to point out that the sample on which it was based can be treated more as a poll. |
Thank you for your valuable insight. I have expanded on the generalizability of this study’s results in the Conclusions section, as shown below:
As far as the research replicability is concerned, this study was aimed at college students, and there were limitations related to the ecological validity of our research results due to the small sample size. Future studies should therefore employ larger research samples, using this article as an introduction for further analysis regarding the process of seeking health information in relation to the COVID-19 pandemic. In terms of the study design, this study asked respondents to answer pre-designed health questions, which may have limited its intrinsic validity, failing to assess the online health information retrieval behavior of individuals when they face personal health problems. In addition to designing a series of health questions to explore the participants’ online health information retrieval practices through observational methods, future research could ask participants to describe their online health information retrieval process in a “think aloud” manner to better understand their subjective use. (lines 334-345)
|
Round 2
Reviewer 1 Report
The proposal has improved. I comment questions:
-Theoretical framework: updated references.
-Methods. You can complete Methodology with a Delphi.
-Results: it will be improved with the Delphi results.
Author Response
Dear Editor:
Thank you for recommending my paper for publication as subject to revision. The paper has been revised according to the reviewers’ constructive comments (see the annotated table below). A number of specific comments and editorial suggestions that were offered by the reviewers and the editor have been incorporated and addressed in the revised paper. In addition, professional editors have edited the paper.
Again, I thank you for kindly offering me this opportunity. I sincerely hope the revised paper is suitable for publication in International Journal of Environmental Research and Public Health. Your comments are greatly appreciated, and I look forward to your reply.
Regards
…..
Wan-Chen Hsu
|
Reviewer’s comments |
Reply |
|
Reviewer 1 |
|
|
1. The proposal has improved. I comment questions:
-Theoretical framework: updated references.
|
Thank you for the comments. I have clarified the theoretical framework and updated the references according to the newly added sources, as below: 1. Theoretical framework Regarding health information retrieval and health promotion theories, the social cognitive theory is one of the most widely used theoretical frameworks [10]. Bandura’s social cognitive theory provides a structure to interpret the relevant results of individuals after retrieving information [11,12]. For example, how much confidence an individual has in finding quality health information, that is, their self-efficacy in searching, is also related to the expected results after retrieval. Self-efficacy can be a powerful predictor of expected results regarding an individual's online health information-seeking behavior [13]. The Risk Information Seeking and Processing Model (RISP) is one of the representative theoretical models explaining online information seeking. It emphasizes that the behavior of individuals to retrieve online information is triggered by insufficient cognitive data (termed as information insufficiency hereafter); according to the model, a lack of information is the main factor directly driving information seeking, alongside other incidental social and psychological factors, such as emotional response (worry, anxiety) and subjective criticism of information. The RISP model thus provides a framework to explain the key influencing factors that individuals use to seek and process relevant risk information in a more systematic or deliberate manner. Brown, Skelly, and Chew-Graham [14] proposed a model, pointing out that individuals’ online health information retrieval is affected by their previous experience, health beliefs, and other personal background factors. (lines 67-84)
The academic contributions of this study could enrich our knowledge and theoretical scope of online health information-seeking behavior issues, highlighting its value for students in the COVID-19 era. (lines 104-106)
2. Newly added references
Bandura, A. The self system in reciprocal determinism. Am Psychol 1978, 33, 344-358. https://psycnet.apa.org/doi/10.1037/0003-066X.33.4.344 Bandura, A. Self-efficacy: The exercise of control. W. H. Freeman and Company: New York, 1997. Brown, R.J.; Skelly, N.; Chew-Graham, C.A. Online health research and health anxiety: A systematic review and conceptual integration. Clin. Psychol. (New York) 2019, 27, e12299. https://doi.org/10.1111/cpsp.12299 Dawkins-Moultin, M.; Mcdonald, A.; Mckyer, E.L. Integrating the principles of socioecology and critical pedagogy for health promotion health literacy interventions. J. Health Commun. 2016, 21, 30-35. https://doi.org/10.1080/10810730.2016.1196273 Gallo, G.; Grossi, U.; Sturiale, A.; Di Tanna, G.L.; Picciariello, A.; Pillon, S.; Mascagni, D.; Altomare, D.F.; Naldini, G.; Perinotti, R.; Telemedicine in Proctology Italian Working Group. E-consensus on telemedicine in proctology: A RAND/UCLA-modified study. Surgery 2021, 170, 405–411. https://doi.org/10.1016/j.surg.2021.01.049 Hollander, J.E.; Carr, B.G. Virtually perfect? Telemedicine for COVID-19. N Engl J Med 2020, 382, 1679–1681. https://doi.org/10.1056/NEJMp2003539 Myrick, J.G. The role of emotions and social cognitive variables in online health information seeking processes and effects. Comput. Hum. Behav. 2017, 68, 422-433. https://doi.org/10.1016/j.chb.2016.11.071 Niederberger, M.; Spranger, J. Delphi Technique in Health Sciences: A map. Front. Public Health 2020, 8, 457. doi: 10.3389/fpubh.2020.00457 Taylor, E. We agree, don’t we? The Delphi method for health environments research. HERD 2020, 13, 11-23. https://doi.org/10.1177%2F1937586719887709 |
|
2. -Methods. You can complete Methodology with a Delphi.
|
Thank you for your comments. I have added a section about the Delphi method in the Materials and Methods section accordingly:
In this study, the Delphi method was used for gaining consensus through controlled feedback from a panel—a group made up of experts in the subject. The method is often used when there is limited or conflicting evidence, the participants may be geographically dispersed, and anonymity is desired to control for dominant individuals. The Delphi method consists of panel selection, the development of content surveys, and iterative stages of anonymous responses to gain consensus [23]. The relevance and objectives of Delphi techniques differ among various disciplines. While they are primarily used in the context of technical and natural sciences to analyze future developments, they are also used in health sciences to reach consensus [24]. (lines 110-124) |
|
3. -Results: it will be improved with the Delphi results.
|
Thank you for your suggestion. I have added the Delphi results in the Results section accordingly: In this study, the Delphi method was used to encode the data content of the respondent's response to the survey results, and use descriptive statistics to restore the current status of college students' online information retrieval behavior. (lines 191-193) |
Reviewer 2 Report
Thanks for making all the suggested changes. Unfortunately the order of the references is wrong (27 in the references section and 29 in the text, do not match)
Author Response
Dear Editor:
Thank you for recommending my paper for publication as subject to revision. The paper has been revised according to the reviewers’ constructive comments (see the annotated table below). A number of specific comments and editorial suggestions that were offered by the reviewers and the editor have been incorporated and addressed in the revised paper. In addition, professional editors have edited the paper.
Again, I thank you for kindly offering me this opportunity. I sincerely hope the revised paper is suitable for publication in International Journal of Environmental Research and Public Health. Your comments are greatly appreciated, and I look forward to your reply.
Regards
…..
Wan-Chen Hsu
|
Reviewer’s comments |
Reply |
|
Reviewer 2 |
|
|
1. Thanks for making all the suggested changes. Unfortunately the order of the references is wrong (27 in the references section and 29 in the text, do not match)
|
Thank you for the comments. I have checked the references and updated the in-text citations as well as the list in accordance with the newly added sources. |